

# A glacial survivor of the alpine Mediterranean region: phylogenetic and phylogeographic insights into *Silene ciliata* Pourr. (Caryophyllaceae)

Ifigeneia Kyrkou[1,2], José María Iriondo[2] and Alfredo García-Fernández[2]

[1] Department of Biotechnology, Agricultural University of Athens, Athens, Greece
[2] Area de Biodiversidad y Conservación, Universidad Rey Juan Carlos, Móstoles, Madrid, Spain

## ABSTRACT

*Silene ciliata* Pourr. (Caryophyllaceae) is a species with a highly disjunct distribution which inhabits the alpine mountains of the Mediterranean Basin. We investigated the phylogeny and phylogeography of the species to (a) clarify the long-suggested division of *S. ciliata* into two subspecies, (b) evaluate its phylogenetic origin and (c) assess whether the species' diversification patterns were affected by the Mediterranean relief. For this purpose, we collected DNA from 25 populations of the species that inhabit the mountains of Portugal, Spain, France, Italy, former Yugoslav Republic of Macedonia, Bulgaria and Greece and studied the plastid regions *rbcL*, *rps16* and *trnL*. Major intraspecific variation was supported by all analyses, while the possibility of the existence of more varieties or subspecies was not favoured. Plastid DNA (cpDNA) evidence was in accordance with the division of *S. ciliata* into the two subspecies, one spreading west (Iberian Peninsula and Central Massif) and the other east of the Alps region (Italian and Balkan Peninsula). This study proposes that the species' geographically disconnected distribution has probably derived from vicariance processes and from the Alps acting as a barrier to the species' dispersal. The monophyletic origin of the species is highly supported. cpDNA patterns were shown independent of the chromosome evolution in the populations and could have resulted from a combination of geographic factors providing links and barriers, climatic adversities and evolutionary processes that took place during Quaternary glaciations.

# INTRODUCTION

Alpine environments provide interesting frameworks for answering phylogeographic and phylogenetic questions that remain unresolved from a botanical perspective. Plant species in mountain ecosystems face challenges for survival and adaptation to different environmental conditions and fluctuations (*Körner, 2003*). High altitude habitats often follow an island-like structure due to significant levels of isolation and fragmentation (*Pawłowski, 1970*), thus leading to adaptive divergence and, finally, speciation events

Corresponding author
Ifigeneia Kyrkou,
ifigeneia.kyrkou@gmail.com

(*Wiens, 2004*). These inland habitat patches could harbour greater species diversity compared to a seamless area of the same extent (*Quinn & Harrison, 1988*). Nunataks and peripheral glacial refugia inside mountain ranges are thought to have sheltered a wide range of biological and genetic diversity during the Pleistocene glacial-interglacial periods (*Hewitt, 2000*; *Taberlet et al., 1998*).

Various phylogeographic and phylogenetic surveys have been conducted for floristic taxa of the Alps (*Schönswetter et al., 2005*), while the rest of the European mountain ranges and the processes occurring inside them during glaciations have generally been overlooked (*Hewitt, 2001*). Nevertheless, interest in Mediterranean mountain systems has gradually been increasing (e.g., *Vargas, 2003*; *Mas de Xaxars et al., 2015*). The Mediterranean Basin has undoubtedly played a crucial role in shaping the genetic and distributional patterns of many species, since it provided them with sanctuary during glaciations (*Médail & Diadema, 2009*) and served as a starting point for the recolonization of northern latitudes (*Petit et al., 2003*; *Tzedakis et al., 2002*). Indeed, the Southern Mediterranean Peninsulas (i.e., Iberian, Italian and Balkan) are considered important glacial refugia for many plant and animal species (e.g., *Taberlet et al., 1998*; *Hewitt, 2000*; *Hewitt, 2004*), and Mediterranean mountains have been considered potential refugia for alpine plants (*Vargas, 2003*; *Hughes, Woodward & Gibbard, 2006*).

*Silene* L. is a genus that has caught the attention of many scientists due to its many interesting attributes, making it a potential "model system" in ecology and evolution (*Bernasconi et al., 2009*). Yet, its phylogeny still remains perplexing and unclear (*Oxelman et al., 2000*; *Greenberg & Donoghue, 2011*). Half of *Silene* species inhabit the Mediterranean Basin (*Greuter, 1995*) and c. 87 of them are found in altitudes above the treeline (based on *Jalas & Suominen, 1988* and supported by *Zángheri & Brilli-Cattarini, 1976*; *Castroviejo et al., 1986–2001*; *Strid & Tan, 2002*). The majority of *Silene* species are diploid with $2n = 20$ or $2n = 24$ (*Bari, 1973*). The latest taxonomic classification can be found in *Greenberg & Donoghue (2011)*. Many recent studies have tried to clarify the phylogeny of its tribes and sections (e.g., *Oxelman et al., 2000*; *Rautenberg et al., 2008*; *Rautenberg et al., 2010*).

Although *Silene* species in alpine environments have been included in phylogenetic and phylogeographic studies of the genus *Silene* (e.g., *Abbott et al., 1995*; *Popp et al., 2005*), those native to Mediterranean mountains have been understudied. *Silene ciliata* is a notable species in the genus *Silene*, because it presents a circum-mediterranean distribution around mountain ranges and above the treeline. Taxonomists have consistently divided it into two subspecies based on habit differences and disjunct geographical distribution. These are *S. ciliata* subsp. *graefferi* (referred to as the "Italian race"), which is principally found in the Italian and the Balkan Peninsula, and *S. ciliata* subsp. *ciliata*, (referred to as the "Spanish race"), which occupies the Iberian Peninsula (*Blackburn, 1933*). *Blackburn (1933)* recorded large morphological and cytological variation both between and within the two races. She concluded that the prime differences inside the "Italian race" are size, leaf form, hairiness and flower colour, whereas variation in the "Spanish race" is unravelled in all features of the plant. For the western populations several other subspecies or varieties have long been proposed (e.g., *Silene ciliata* subsp *arvatica* Lag. in Varied .Ci. (1805),

*Silene ciliata* subsp. *elegans* (Link. ex Brot.) Rivas Martínez in *Brotero, 1804*), although the validation of these subcategories remains unsolved with available taxonomical data (*Nieto Feliner, 1985*). This species also stands out for its extraordinary variability of ploidy levels in natural populations (i.e., $2n = 24, 36, 48, 72, 84, 96, 120, 144, 168, 192, 240$; *Blackburn, 1933*; *Küpfer, 1974*). In particular, subsp. *ciliata* is reported to vary from diploid to 20-ploid complements, whereas in subsp. *graefferi* only diploid and tetraploid plants are described (*Blackburn, 1933*; *Küpfer, 1974*; *Tutin et al., 1993*).

We followed a phylogenetic and phylogeographic approach to this species to gain insight into the diversification processes that have taken place in alpine environments of Mediterranean high mountains. To our knowledge, this is the first study to cover the vast majority of the alpine Mediterranean area with the aid of molecular marker evidence. We hypothesize that: (1) in spite of its heterogeneity discussed by Blackburn in *1933*, the species is of monophyletic origin; (2) this heterogeneity is reflected in great cpDNA diversification that could explain the sub-classification of this species into two distinct subspecies as proposed by *Blackburn (1933)* and maintained by *Tutin et al. (1993)* and; (3) differentiation patterns are essentially determined by the geomorphology and spatial location of the Mediterranean mountain ranges.

## MATERIAL AND METHODS

### Studied species

*Silene ciliata* Pourr. (subsect. *Fruticulosae*, Caryophyllaceae) is endemic to Europe and inhabits the main Mediterranean mountain ranges in the northern half of Mediterranean Basin countries spreading along the Iberian Peninsula, the Central Massif, the Apennines and the Balkan Peninsula (*Tutin et al., 1993*). It is an alpine, chamaephytic, perennial, cushion plant, which typically forms pulviniform rosettes of up to 2 cm in height and 15 cm in diameter with high variability in size. Each plant has an average of $13 \pm 11$ cm (mean $\pm$ SD) flowering stems that reach 15 cm in height and bear 1–5 flowers (*Giménez-Benavides, Escudero & Iriondo, 2007*).

### Taxon selection

Twenty-five specimens of *S. ciliata* populations covering the species distribution range were sampled for this study (Fig. 1). Plant material was obtained from herbarium specimens or directly from the field and stored as silica gel-dried material (Table 1). All field studies carried out by the authors were conducted with the permission of "Junta de Castilla y León" and "Comunidad de Madrid" (approval code numbers: 20144360000894 and 10/117476.9/14, respectively). To assess possible intrapopulation cpDNA variation, DNA from four additional individuals of the Cen3 population was also extracted and amplified.

For the estimation of the phylogeny of the polymorphic cpDNA region, eight additional species of genus *Silene*, phylogenetically close to *Silene ciliata*, were included in the study. The selection of these species was based on the most recent phylogenetic studies of *Sileneae* (*Sloan et al., 2009*; *Greenberg & Donoghue, 2011*) and the availability of the required polymorphic cpDNA regions. The search was performed in the GenBank sequence

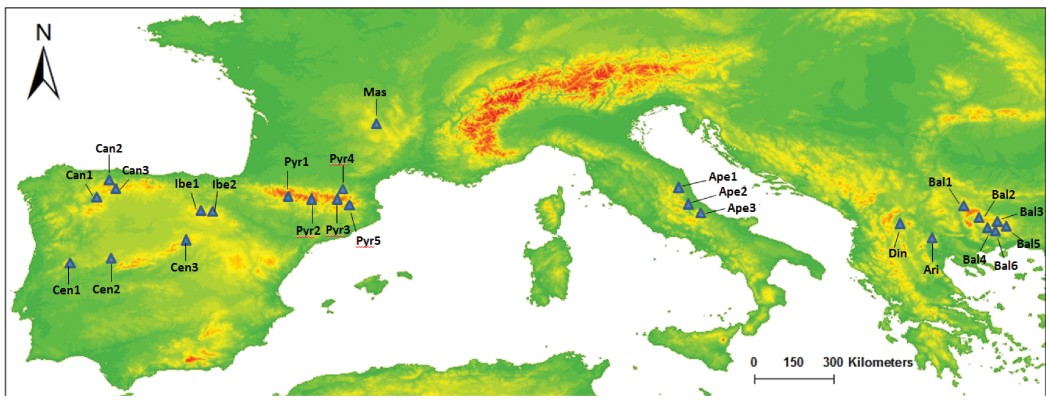

**Figure 1 Map of our sampled populations of *Silene ciliata*.** Distribution of sampled *S. ciliata* populations in the Mediterranean Basin. Acronyms were derived from the name of the mountain system where samples were collected: Can, Cantabrian Range; Ibe, Iberian System; Pyr, Pyrenees range; Cen, Central System; Mas, Central Massif; Ari, Aridaia range; Bal, Balkan-Rhodope mountain system; Din, Dinaric Alps; Ape, Apennines range.

database, and the species selected as outgroups were *S. latifolia* Poiret, *S. uniflora* Roth, *S. vulgaris* (Moench) Garcke and -phylogenetically closer to *S. ciliata*—*S. acaulis* (L.) Jacq, *S. otites* (L.) Wibel, *S. nutans* L., *S. paradoxa* L. and *S. schafta* S. G. Gmel. ex Hohen. Although *S. borderei* Jordan and *S. legionensis* Lag. are classified in the same section as *S. ciliata* in recent Floras (e.g., Flora Europaea, *Tutin et al., 1993*; Flora Iberica; *Castroviejo et al., 1986–2001*), we did not consider them because they are not included in the phylogenetic study of *Sileneae* by *Greenberg & Donoghue (2011)*. The accession numbers of all outgroup regions are listed in Table S3.

## DNA markers

In this phylogenetic and phylogeographic approach, we chose maternally inherited plastid DNA (hereafter cpDNA) as the focus of our study, since it provides a conservative and enduring record of plant migrational spread (*McCauley, 1997*; *Irwin, 2002*) compared to biparentally inherited nuclear markers that show recombination (*Petit, Kremer & Wagner, 1993*; *Heuertz et al., 2004*).

Each of the 25 extracted DNA samples was amplified for the *rbcL*, *rps16* and *trnL* polymorphic cpDNA regions. These regions were selected from the 12 regions that had previously shown major variation and the best amplification profile (*Shaw et al., 2005*; *Shaw et al., 2007*).

## DNA extraction and amplification

For DNA extraction, approximately 20 mg of dried leaf tissue of each plant sample were weighed. Extractions were performed following the protocol of Qiagen Plant DNA extraction kit (QIAGEN Inc., CA, USA) with some modifications. The primers used and the PCR conditions applied for each marker, as well as the primer sequences and references, are listed in Table S2. The PCR mix was prepared using PureTaq Ready-To-Go PCR beads (GE Healthcare, Uppsala, Sweden).

Table 1 **Details of the sampled populations of *Silene ciliata*.** DNA samples of *Silene ciliata* used for the study. The table shows the acronym given to each sampled population ("Name"), the "Country" where these populations were collected, "Altitude" and MGRS coordinates. A more detailed version of this table can be found in Table S1.

| Name | Country | Altitude (m) | MGRS |
| --- | --- | --- | --- |
| Can1 | ES | 1,642 | 29TQH4477 |
| Can2 | ES | 1,900 | 30TUN3712 |
| Can3 | ES | 1,881 | 30TUN5150 |
| Ibe1 | ES | 1,900 | 30TVM9646 |
| Ibe2 | ES | 2,278 | 30TWM0276 |
| Pyr1 | ES | 1,931 | 30TYN2920 |
| Pyr2 | ES | 1,350–1,780 | 30TYN4026 |
| Pyr3 | ES | 2,100–2,200 | 31T CG7967 |
| Pyr5 | ES | 2,161 | 31TDG1980 |
| Cen2 | ES | 1,950 | 30TTK7079 |
| Cen3 | ES | 2,340 | 30TVL2104 |
| Cen1 | POR | 1,900 | 29TPE1783 |
| Mas | FR | 1,560 | 31TDL8119 |
| Pyr4 | FR | 2,190 | 31TDH3461 |
| Ari | GR | 2,182 | 34TFL0142 |
| Bal3 | GR | 1,800 | 35TKF5580 |
| Bal4 | GR | 1,800 | 35TKF5307 |
| Bal5 | GR | 1,800 | 35TKF5586 |
| Bal6 | GR | 2,060 | 35TKF5632 |
| Bal1 | BU | 1,900 | 34TGM0365 |
| Bal2 | BU | 2,600 | 34TGM0229 |
| Din | MAC | 2,480 | 34TEM2771 |
| Ape1 | IT | 1,950 | 33TUH8528 |
| Ape2 | IT | 1,366 | 33TUH7979 |
| Ape3 | IT | 2,000 | 33TVG2225 |

## Data analyses

Sequencing results were evaluated and corrected manually before being subjected to multiple alignment. The manual corrections were made to check whether the differences found among some bases of the sequences were unique/repeated in some of the sequences and to ensure the presence of gaps. Contigs were assembled and edited with Sequencher 4.1.4 (Gene Codes Corp., MI, USA), Bioedit (*Hall, 1999*) and ClustalW (*Thompson, Higgins & Gibson, 1994*). In the latter, default settings were used.

The number of variation and informative sites of our aligned sequences was determined using DnaSP v.5.10.01 (*Librado & Rozas, 2009*). The phylogenetic analyses were performed using two different statistical approaches ("Bayesian inference" and "Maximum likelihood") for verification reasons. In the Bayesian analysis, sequence data were first introduced to jModeltest (*Posada, 2008*) to determine the best fitting evolutionary model according to the AIC criterion. This process was followed to generate a dendrogram for

each polymorphic cpDNA region, plus one dendrogram that included all polymorphic cpDNA regions together. The suggested model for *rbcL* was [HKY], for *rps16* [GTR + G], for *trnL* [HKY + I] and for the tree including all markers [GTR + G]. These models were then inserted into MrBayes 3.1.2 (*Huelsenbeck et al., 2001*) and posterior probabilities (hereafter PP) were estimated using the Markov chain Monte Carlo (MCMC) method. Four Markov chains were run in parallel for 10,000,000 generations and sampled every 100 generations. The first 100 generations were set as the "burn-in" period, while the rest were used to calculate the 50% majority rule consensus phylogeny and posterior probability. The resulting dendrogram archives were revised with FigTree v. 1.3.1 (*Rambaut, 2009*). A maximum likelihood dendrogram including all the polymorphic cpDNA regions together was also generated with PhyML 3.0 (*Guindon et al., 2010*) under the same evolutionary model used for the Bayesian analysis. The reliability of the branches was calculated through bootstrapping, after producing 1,000 bootstrapped data sets. All outputs were compared and analysed to infer the evolutionary history of our study species.

Next, each group of polymorphic cpDNA region sequences was analysed with TCS 1.2.1 (*Clement, Posada & Crandall, 2000*) and classified according to statistically parsimonious haplotype groups. The haplotype groups were linked by the program, constructing a network of mutation steps, which visualized the genetic distance between them. For the construction of the haplotype networks, deletions were not treated as polymorphic sites, while the analysis was performed under the default of 95% connection limit. To facilitate interpretation, a total cpDNA haplotype network was created with this method. In addition, haplotype networks of *rbcL*, *rps16* and *trnL* regions were obtained separately to check the congruence between markers. Likewise, total neighbour-net analysis network including all three cpDNA regions together was also designed using Splits Tree v. 4.13.1 (*Huson & Bryant, 2006*) and following the uncorrected p-distance between individuals. The support for each branch was tested using the bootstrapping method with 1,000 replicates. One final test was performed with the Bayesian Analysis of Population Structure 6 (BAPS, *Corander et al., 2008*). This Bayesian approach is conditioned on the geographical sampling information available. The actual analysis is performed using a systematic hierarchical Bayesian approach, where a Markov chain Monte Carlo (MCMC) estimation is used whenever the number of possible partitions is too large to be handled with exact calculations. We chose BAPS software to infer the best genetic structure, considering the coordinates of each sample, and ran a test of spatial clustering of individuals, with five replicates for each possible number of groups (*K*).

## RESULTS

### Chloroplast haplotype and intrapopulation variation

After multiple alignment evaluation of the three polymorphic cpDNA regions, the final length of the study region resulted in 564 nucleotides for *rbcL*, 756 nucleotides for *rps16* and 509 nucleotides for *trnL*. Thus, the length of the combined matrix of an "all-marker" region was 1,829 nucleotides. The number of variable sites among chloroplast markers ranged from 4 to 25, while that of parsimony informative sites ranged from 3 to 16
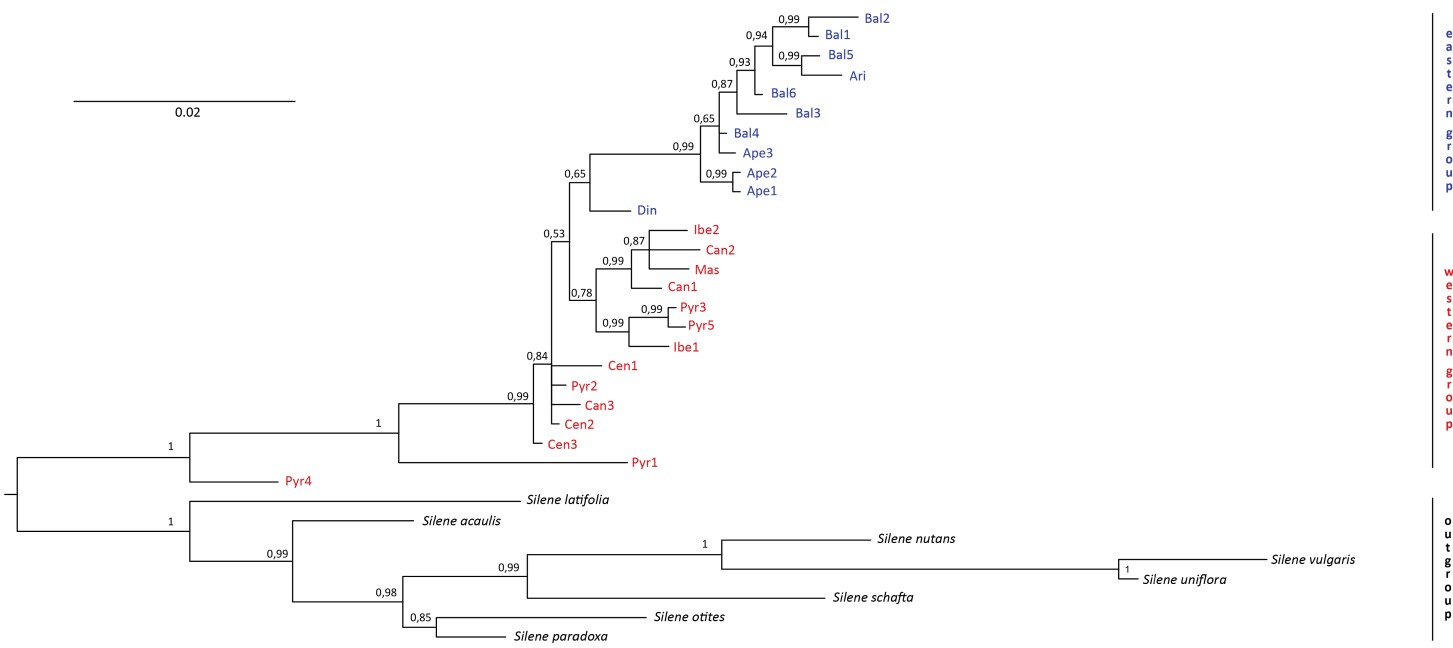

**Figure 2 Bayesian dendrogram.** Bayesian consensus dendrogram of the "all-marker" cpDNA sequence of *Silene ciliata*.

**Table 2 Characteristics of the polymorphic cpDNA regions.** Characteristics of the three polymorphic cpDNA regions and the "all-marker" region studied in *Silene ciliata*. The length of the products after amplification with the corresponding marker and alignment editing, and the variable and parsimony sites of each product ensued from the DnaSP analysis are shown.

| Chloroplast marker | Length of selected region | Variable (polymorphic) sites | Parsimony informative sites |
|---|---|---|---|
| *rbcL* | 564 bp | 4 | 3 |
| *rps16* | 753 bp | 25 | 16 |
| *trnL* | 513 bp | 18 | 11 |
| all | 1,830 bp | 47 | 30 |

(Table 2). Sequences were submitted to GenBank (accession numbers are available in Table S4).

The intrapopulation study showed no divergence for *rbcL* and inconsistent polymorphisms (i.e., only present in one individual and probably associated to sequencing errors) in one and two bases for *rps16* and *trnL*, respectively. Therefore, we considered that the evidence for intrapopulation variation was not strong enough to require further testing.

## Phylogeny, genetic distance analyses and population structure

No incongruence in results was found among the single markers. Therefore, we essentially used a combined study because all the markers are in the chloroplast genome. The resulting "all-marker" dendrogram from the Bayesian analysis (Fig. 2) revealed two distinct groups, one including all individuals in the western region (i.e., the Iberian Peninsula and France)

and another one including all individuals in the eastern region (i.e., the Italian and Balkan Peninsulas). However, the calculated 65% PP for the "eastern group" did not provide a significant difference between the two groups. On the other hand, significant differentiation (100% PP) was found between *S. ciliata* individuals and the outgroups. Strikingly, two *S. ciliata* individuals, Pyr1 and Pyr4, were located between the outgroups and the rest of *S. ciliata*, and were significantly different from them as well as from each other. Both Pyr1 and Pyr4 branches were long, implying high substitution rates. One overarching clade was observed (99% PP) in the "eastern group", and the Din population was the only one branching off this clade. The "western group" consisted of one clade (78% PP), but also had many separate individual branches. The maximum likelihood dendrogram obtained with the bootstrapping method did not differ, either in formation or in significance of branches support, from the Bayesian dendrogram.

In the haplotype network approach, the overall analysis (Fig. 3) corroborated the existence of two (eastern–western) groups and found higher haplotype diversification in the western group. Of all the eastern populations, the Din haplotype had the nearest position to the western group, which was in agreement with the results obtained from the dendrogram. Sequences assembled into 24 haplotypes, with 12 haplotypes including only western region sequences, 10 haplotypes including only eastern region sequences and two haplotypes outside the network (Pyr1 and Pyr4). Only eastern region sequences Ape1 and Ape2 shared the same haplotype pattern, and no shared haplotype patterns were found between the "eastern" and "western" groups. Moreover, the haplotype network revealed a close relationship between the haplotype pattern of Ape3 and some Balkan populations and among the haplotype pattern of Pyr2 and some Central System populations. The *rbcL* haplotype network (Fig. S1) was selected to visualise the geographic distribution of haplotypes by regions, as it showed the most representative and parsimonious patterns of the three cpDNA regions, when analysed separately (Fig. 5). In that network, Cen2 and Bal1 haplotypes were prevalent in the western and eastern regions, respectively.

The neighbour-net method suggested a grouping pattern that was in accordance with the one obtained using the haplotype network approach. Besides that, it provided a chance to delve deeper into the differences among *S. ciliata* sampled populations. The all-inclusive neighbour-net network (Fig. 4) confirmed the classification of all studied populations into a western and an eastern region, which was 91.6% statistically supported. Furthermore, some distances inside the network were noteworthy because they verified previous results. This is the case of the observed 98.4% difference in the distance between Cen1 and the rest of Central System populations (implied by the haplotype network). The Italian Ape3 showed a minor differentiation (77.1%) that was also noticed in the haplotype network and in the dendrograms. Last but not least, the eastern population Din was placed in the "western" group.

The Bayesian spatial clustering of populations resulted in an optimal grouping of $K = 2$. This supported the western–eastern region division of populations noted in previous analyses. Only the Balkan population Din deviated once more from this division, clustering with the western-region populations.

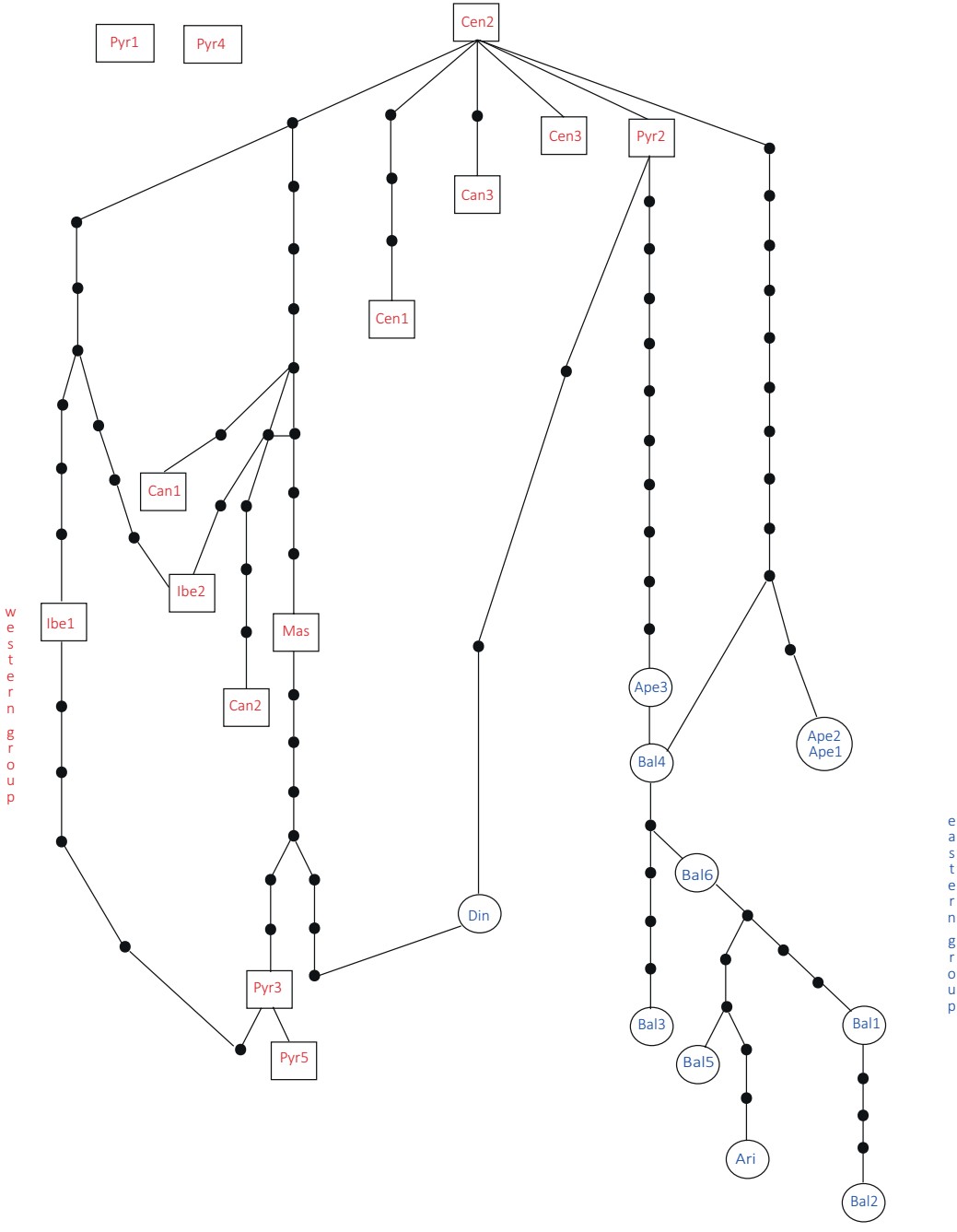

**Figure 3 Total cpDNA haplotype network.** Combined haplotype network analysis including all cpDNA markers and showing the relationships between the cpDNA parsimony haplotype groups. Rectangles and ovals depict haplotypes that belong to the western and eastern groups, respectively. The patterns of individuals Pyr1 and Pyr4 are segregating from the rest of the haplotypes.

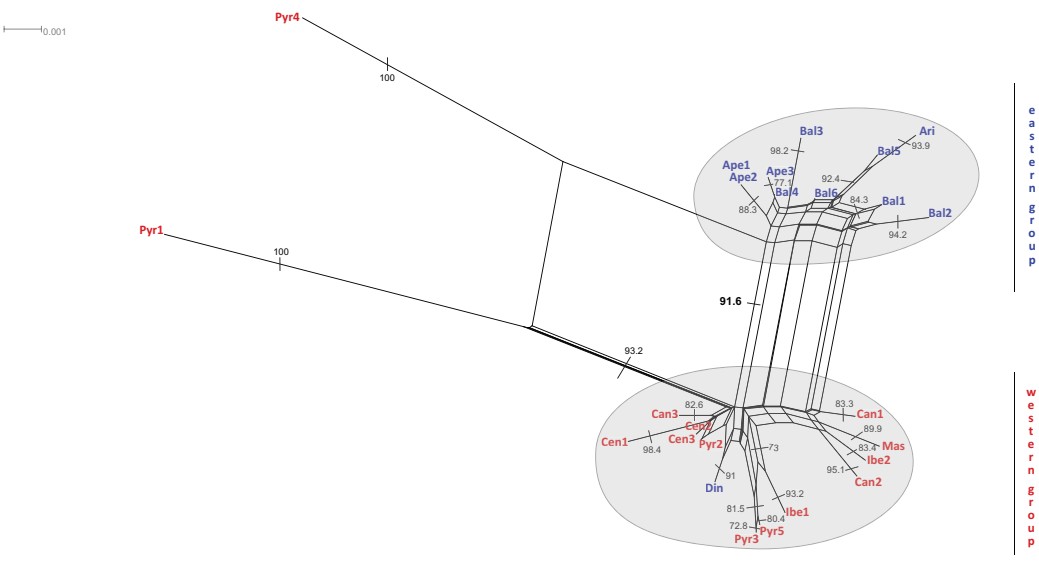

**Figure 4 Neighbour-net analyses of the total matrix.** Neighbour-net analyses of rbcL (A), rps16 (B) and trnL (C) based on uncorrected *p*-distances. Numbers denote significant bootstrapping values. The eastern and western groups of *S. ciliata* populations are indicated by grey-shaded clusters. Blue letters correspond to the eastern group and red letters to the western group.

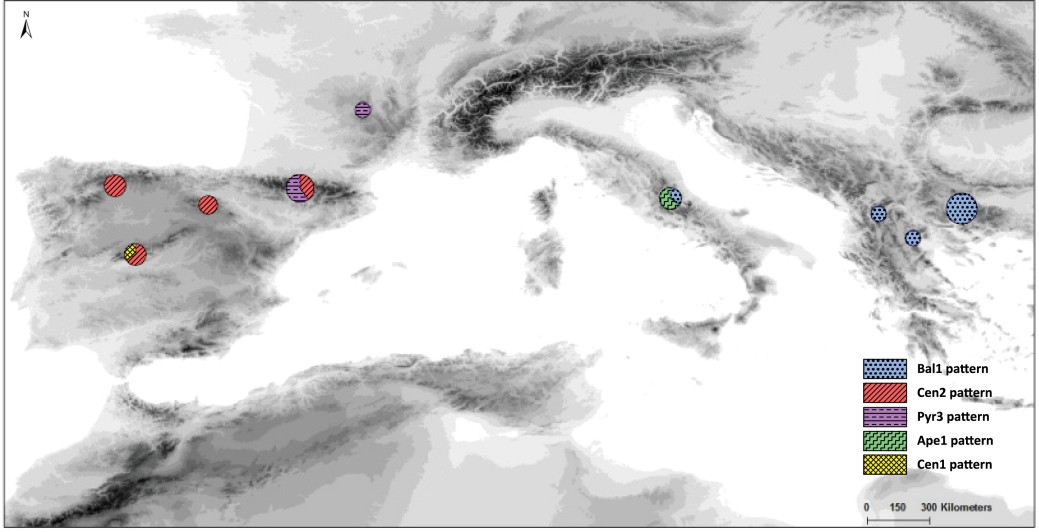

**Figure 5 Distribution and frequency ratios of *rbcL* haplotypes.** Distribution and frequency ratios of *S. ciliata* haplotypes for *rbcL* (see Fig. 4A) in the mountain systems of this study. The proportion of different haplotypes at each location is shown in the circles.

## DISCUSSION

### Genetic diversity in the cpDNA of *S. ciliata*: a comparative approach

This study reveals high haplotype variability and therefore supports the hypothesis of high cpDNA diversification among *S. ciliata* populations. Similar results have been reported

in previous studies on other *Silene* species, such as *S. latifolia* (*Ingvarsson & Taylor, 2002*), *S. vulgaris* (*Štorchová & Olson, 2004*) and *S. dioica* (*Hathaway, Malm & Prentice, 2009*), among others. Yet, *S. ciliata* is ranked among the most varied. Low levels of cpDNA diversification and no diversification at all have been found in *S. hifacensis* (*Prentice et al., 2003*) and *S. sennenii* (*López-Vinyallonga et al., 2012*), respectively, although this may be due to their narrow distributions and low population sizes (*Gitzendanner & Soltis, 2000*; *López-Pujol et al., 2009*; *López-Vinyallonga et al., 2012*). Considering this, we suggest that the variation detected in *S. ciliata* is the outcome of a relatively ancient, wider distribution range, followed by a gradual splintering caused by a series of ice ages, as proposed for many other high-elevation species (reviewed by *Nieto Feliner, 2014*). An alternative explanation to vicariance is that diversification patterns resulted from dispersal. However, long-distance dispersal events most likely played a minor role given that *S. ciliata* seeds lack any specialized dispersal structure and, thus, most seeds are barochorously dispersed at distances of less than 1 m (*Lara-Romero et al., 2014*). Finally, although the step node of the species is dated around 10 million years ago (*Sloan et al., 2009*), and the diversification of the southern European alpine flora has been correlated with the climatic oscillations of the Pleistocene (*Kadereit, Griebeler & Comes, 2004*), we have no information about the date of the crown node. Therefore, we have no evidence of the period when the diversification into the two subspecies took place.

## Interpreting the distinction of *S. ciliata* between western and eastern regions and their origin

No evidence was found against the classification of *S. ciliata* into a western and an eastern race (*Blackburn, 1933*; *Tutin et al., 1993*). Hence, we propose maintaining the names *Silene ciliata* subsp. *ciliata* and *S. ciliata* subsp. *graefferi* to describe the noted clustering of *S. ciliata* individuals into a western and eastern group, respectively. On the other hand, both dendrograms indicated a significant difference between *S. ciliata* individuals and the outgroups which, together with the nonessential divergence between populations, corroborates the monophyly of our species.

Tracing back to the species' differentiation, we hypothesize that populations of an ancestor of *S. ciliata* dominated the Mediterranean Basin. At the onset of glacial period climatic oscillations in the late Tertiary and in the Quaternary period, these ancestral populations might have been forced to migrate to favourable areas, while those unable to encounter a glacial refugium because of distance, time or natural barriers perished. Given that we are dealing with an alpine species, *S. ciliata* populations could have migrated following the paths that constitute links between neighbouring mountains. The Alps mountain range system seems to have posed a persistent and significant hurdle for this species' migration. A rigorous example supporting this theory is that during Quaternary glaciations, the Alps, in contrast to the Mediterranean mountains, were extensively covered with ice sheets (*Hughes, Woodward & Gibbard, 2006*; but see *Stehlik et al., 2002*). This is in accordance with previous phylogeographic studies (e.g., *Taberlet et al., 1998*; *Hewitt, 2000*) and may explain why *S. ciliata* populations have not been found there. Moreover,

it would account for the observed disconnected distribution and division of the species into the western and eastern groups, since the geographical borders formed by the two groups coincide with the location of the Alps. A similar grouping pattern has been found in the Mediterranean for *Androsace vitaliana* (*Vargas, 2003*) and *Heliosperma* (*Frajman & Oxelman, 2007*), genera with the barrier shifting west and east of the Alps region, respectively. Disjunction in distribution, possibly resulting from the Alps and distinction into two subspecies has recently been proposed in *Artemisia eriantha*, another alpine plant distributed along the Alps and many Mediterranean mountains (*Sanz et al., 2014*), which also supports our hypothesis.

## Evolutionary processes and geo-climatic effects on western and eastern populations

Apart from the significant difference found between eastern and western cpDNA sequences, further important diversification was found inside each group. Polyploidization is an evolutionary mechanism that has generated evolutionary lineages during the Pleistocene (*Stebbins, 1984*). The number of chromosomes of *S. ciliata* is mostly $n = 24$ or $n = 48$, although populations with much higher levels of ploidy have been described in *S. ciliata* subsp. *ciliata* (*Blackburn, 1933*; *Küpfer, 1974*; *Tutin et al., 1993*). Hence, we propose that this mechanism could explain some of the differentiation within *S. ciliata*. Nevertheless, considering available chromosome data, we found no relationship between chromosome number and the clustering obtained in our results. Variation may have also resulted from slow mutation events inside disjunct refugia during periods of adverse climatic conditions (*Sanz et al., 2014*) either as an outcome of elevational range shifts (lowland glacial refugia) or *in situ* endurance (nunataks). Additionally, other sources of diversification like genetic drift associated to low population sizes and prolonged isolation should be considered (*Young, Boyle & Brown, 1996*).

Regarding the western group, genetic diversity is apparent in the Pyrenees mountain range and has led to the genetic disaffiliation of the range into a western and an eastern section. This is in line with the genetic break found in *Artemisia eriantha* (*Sanz et al., 2014*). Another component of the western group diversification was introduced by the highly divergent Cen1 sequence of Serra da Estrela, suggesting high isolation of this population. On the other hand, the merging of Pyr2 sequence with Central System *S. ciliata* individuals may imply braided migrational paths between these populations during glacial-interglacial events.

Interestingly, the degree of divergence recorded in the eastern group of *S. ciliata* is higher than that in the western group. This observation has also been made for temperate trees and shrub taxa (*Petit et al., 2003*). This higher genetic diversity and the existence of more unique haplotypes, especially in the Balkan Peninsula, might be due to the additional effect of the complex orography and restricted territorial extent of existing refugia, which did not facilitate exchange among populations. More specifically, the various orientations of mountain chains in the Balkans may have acted as a barrier to migration (*Tzedakis, 2004*). The individuals from the western part of the eastern groups (e.g., Ari and Din) showed

some important differences in certain analyses (see Figs. 3 and 4). This might be related to the nature of the east Balkan slopes, which have a more gentle relief compared to the steep west mountains (*Reed, Kryštufek & Eastwood, 2004*), thereby fostering higher levels of isolation. Further differentiation in Din could be because the Dinaric Alps were much less affected by glaciations than the rest of the Mediterranean mountain systems (*Frajman & Oxelman, 2007*), resulting in the maintenance of relict populations. Lastly, the close relationship of Italian Ape3 with some Balkan populations (especially Bal4) might result from the proposed land connection of the north Italian and Balkan Peninsulas during the early Holocene (approx. 20–16 ka BP) (*Lambeck et al., 2004*) which would have facilitated dispersal events between the two regions.

## The Pyrenees case

The Bayesian and maximum likelihood analyses showed that Pyr1 and Pyr4 differed from the outgroups as well as from the rest of *S. ciliata* individuals and were situated in an intermediate position between them in the dendrogram (see Fig. 2). Similar results were found in the rest of the analyses. We surmised that this pattern could be another example of the Pyrenees range acting as a stable hybrid zone, as argued in *Chorthipopus parallelus* (*Hewitt, 1993*) and *Saxifraga* subsect. *Triplinervium* (*Mas de Xaxars et al., 2015*). At any rate, the rise of hybrid zones due to glaciations, and hence, the preservation of different species genomic information via gene flow (*Harrison, 1990*) are linked with high altitudes (*Hewitt, 2001* and references therein). In the case of Pyr1 and Pyr4, their haplotype patterns may have resulted from chloroplast capture between *S. ciliata* and other congeneric, sympatric species (*Rieseberg & Soltis, 1991*). After all, the geographical contact of congeneric species causing chloroplast sharing has been reported in other studies including species in the same genus, like *S. latifolia* and *S. dioica* (*Prentice, Malm & Hathaway, 2008*), as well as in other plant groups (e.g., *Gardner et al., 2004*; *Okuyama et al., 2005*). Given the number of haplotype patterns detected inside the species, the alternative explanation of Pyr1 and Pyr4 resulting from lineage sorting is another option that cannot be readily rejected (*Galtier & Daubin, 2008*).

## Conclusions and future prospects

Our results confirm the monophyly of *S. ciliata* due to the differences found between the studied populations and the outgroups and reveal a clear west-to-east division of *S. ciliata* populations with the borderline set in the region of the Alps. This division validates the past classification of the species into two subspecies; *S. ciliata* subsp. *ciliata* found west of the Alps ("Spanish race", *Blackburn, 1933*) and *S. ciliata* subsp. *graefferi* located east of the Alps ("Italian race", *Blackburn, 1933*). Major intraspecific variation is supported by all analyses, but none of them supports the occurrence of additional varieties or subspecies (according to *Küpfer, 1974*; *Castroviejo et al., 1986–2001*). In addition, we suggest that geographic and climatic factors may have played a central role in the evolutionary history of the species and the formation of the two subspecies. Further analyses including more individuals and cpDNA markers, as well as mitochondrial DNA (mtDNA) markers and nuclear ribosomal internal transcribed spacer (nrITS) regions, are encouraged to

secure conclusions of this role and clarify the status of unsolved-incongruent populations. Molecular clocks, the inclusion of additional congeneric species and increased sampling effort are necessary to resolve the remaining questions.

## ACKNOWLEDGEMENTS

We thank the curators of the Muséum National d'Histoire Naturelle (Paris, France), the Natural History Museum (Florence, Italy), Herbarium Apenninicum (Parco Nazionale del Gran Sasso e Monti della Laga, Italy), Herbario de Jaca (Huesca, Spain), Sofia Herbarium (Bulgaria), Herbarium of the Botanical Institute of the University of Patra (Patra, Greece), Herbario de la Universidad de Barcelona, (Barcelona, Spain) and Herbario del Real Jardín Botanico (Madrid, Spain) for the loan of *Silene ciliata* material. We would also like to thank Panayiotis Trigas, Angela López, Marcos Méndez, Luis Giménez-Benavides, Rubén Milla and Alvaro Bueno for plant collection and Lori De Hond for linguistic assistance.

### Funding

This work was supported by the AdAptA project (CGL2012-33528) of the Spanish Ministry of Economy and Competitiveness and an ERASMUS mobility grant. The funders had no role in study design, data collection and analysis, decision to publish, or preparation of the manuscript.

### Grant Disclosures

The following grant information was disclosed by the authors:
AdAptA: CGL2012-33528.
ERASMUS mobility grant.

### Competing Interests

The authors declare there are no competing interests.

### Author Contributions

- Ifigeneia Kyrkou conceived and designed the experiments, performed the experiments, analyzed the data, contributed reagents/materials/analysis tools, wrote the paper, prepared figures and/or tables.
- José María Iriondo conceived and designed the experiments, contributed reagents/materials/analysis tools, reviewed drafts of the paper.
- Alfredo García-Fernández conceived and designed the experiments, performed the experiments, analyzed the data, contributed reagents/materials/analysis tools, prepared figures and/or tables, reviewed drafts of the paper.

### Field Study Permissions

The following information was supplied relating to field study approvals (i.e., approving body and any reference numbers):

The study in the National Park of Sierra de Guadarrama was permitted by Consejería de Medio Ambiente y Ordenación del Territorio of the Autonomous Region of Madrid with approval number 10/117476.9/14. The study in the Regional Park of Sierra de Gredos was permitted by the Delegación Territoria de Ávila, Servicio Territorial de Medio Ambiente of the Committee of Castilla y León with approval number 20144360000894.

## Supplemental Information

Supplemental information for this article can be found online at http://dx.doi.org/10.7717/peerj.1193#supplemental-information.

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
