# Peer review of "A glacial survivor of the alpine Mediterranean region: phylogenetic and phylogeographic insights into Silene ciliata Pourr. (Caryophyllaceae)"

_PeerJ, doi:10.7717/peerj.1193_

## Round 0.1 · original submission · Major Revisions

· Academic Editor

Major Revisions

Dear authors

Thank you for submitting your manuscript to our journal. As you see our reviewers suggest a major revision of your ms. If you are willing to do so, we would be happy to reconsider your revised manuscript.

Michael Wink

·

Basic reporting

The manuscript is written in mostly fluent and unambiguous English. However, it should be carefully checked to conform to professional standards. The background and prior literature is appropriately presented in the introduction. Three regions of the chloroplast genome are used to to a) clarify the suggested division of S. ciliata into two subspecies, b) evaluate its phylogenetic origin and c) assess whether the species´ diversification patterns were affected by the Mediterranean relief. I don't think that any of these questions are unambiguously solved by the data analyses, although there are certainly aspects that are relevant, especially a), and maybe also c).

Experimental design

The description of the multiple alignment procedure does not enable reproduction. Please, describe all parameter settings used in ClustalW. If manual adjustments have been made, please descibe how and why. Include the alignments in supplementary data. Also, please describe the rationale behind the BAPS analysis (for example, which is the grouping criterion?) and the "Voronoi tessellation". Otherwise, the description of methods are fine.
I see no point in analysing the cpDNA regions separately. Usually, the chloroplast genome is assumed to behave as a non-recombining, uni-parentally inherited linkage unit. Therefore, much space can be saved, and clarity achieved, by concatenating the three alignments. Missing data is ok as long as there is overlap.
Vouchers should be deposited in public herbaria.

Validity of the findings

The discussion is ok, but very long. It is not always clear when the discussion has a basis in the results and not.
The conclusion that the cpDNA sampled from S. ciliata is monophyletic is correct given the sampling. However, the authors failed to sample S. borderei Jordan, and S. legionensis Lag., which are classified in the same section in recent Floras (e.g., Flora Europaea, Flora Iberica), and from a morphological and geographical perspective very similar. The non-sampling of these taxa should at least be discussed. The authours should explain how directionality of historical dispersal is inferred from the data.

Comments for the author

Some specific comments:
1. In the introduction it is said that 87 species are found in latitudes above the treeline. Do you mean 87 European species at altitudes above the tree-line?
2. You make heavy use of the Blackburn (1933) refernce when you formulate your hypotheses, so please introduce it more in the introduction
3. Line 231 "High diversification rates" cannot be inferred from the gene tree presented, possibly high substitution rate. Besides, the pattern displayed by Pyr1 and Pyr4 reminds me of patterns you can see if alignments are poor (e.g., one sequence has shifted for a large part), something that often happens when ClustalW alignments are uncritically accepted. I am not saying this is the case, but I suggest that the authors check this, and make the used alignments available.
3. Line 395: cpDNA is maternally inherited and haploid, so hybridization cannot be inferred.
4. Line 420: I disagree that there is any EVIDENCE provided ofthe central role played by geographic and climatic factors

·

Basic reporting

no comments see Comments for the Author.

Experimental design

no comments see Comments for the Author.

Validity of the findings

no comments see Comments for the Author.

Comments for the author

The paper by Kyrkou et al. analysed the phylogenetic relationships of Silene ciliata (= S.c.). They basically found a split between western and central-eastern Mediterranean populations.

I regard the ms. as interesting, the used methods as appropriate and the ms. well-written. Though, I have some major points to consider. 1) The taxon sampling is relatively poor. 2) The drawn conclusions are often too far-reaching. 3) Several things can be shortened.
The first point cannot be changed easily, but it should be stressed that the results are therefore somewhat preliminary. After addressing points 2 & 3 I consider the study as acceptable.

Very positive was the frame of Mediteranean mountains and the inclusion of the relevant literature. The applied methods are appropriate, though I find them rather overdone (see 3).

1) Taxon sampling:
The distribution area of S.c. is quite well covered. Though, including more accessions would have been useful. Especially in the Balkan region some locations are missing. The taxon seems to occur in the area of Mount Olympus. Are there Turkish population? See http://www.tropicos.org/Image/100191204
What was not clear to me if all the described subspecies (arvatica etc.) were included? If so, what about their no. of chromosomes? And the evolution of chromosome no.? This should worked out much better.

Despite several disadvantages nrITS sequences could deliver an independent data set besides cpDNA.

2) Several cases of differentiation within S.c. (l. 335) are too far interpreted. Without a dating approach it is likely that diversification happened in the Pleistocene, but this is a long, very dynamic period. Alternative explanations than e.g,. orographic structures are obvious. See below in each point.

3) The authors should follow a whole-evidence approach by combing all the cpDNA data in single analyses. A short statement that no incongruence was found among the single marker is sufficient. So haplotypes of single markers can be omitted, but instead use a combined study because the markers are all in the cp genome. So instead of figs. 3-5 provide a Neighbour-net analysis of all combined markers. I would only show Haplotype networks if they are informative.

Further places to shorten see below.

Several points:
- p. 1, write out FYROM
- l. 62 this paragraph is superfluous or should be mentioned in MM.
- l. 71-73. I would delete the sentence. The whole block on Silene could be shortened. There is no need for such a justification to study S.c. Just refer to the phylogenetic studies. One can directly state S.c. is one of the Mountain species.
- l. 124-131. Relevant for the study? Delete.
- l. 135. Shorten paragraph. PCR and sequencing is standard and can be described in less words.
- l. 158 => taxon selection
- l. 298. Difficult to argue like this. Even if the stem node may be 10 my old (Sloan et al. 2009), you don’t know the crown node of S.c. which is likely much younger than that.
- l. 320. Completely? But Stehlik et al. 2002 found nunataks in the Alps for Eritrichium nanum.
- l. 350-361 I find highly speculative. The relationships are not well supported. Linking them to single events in the Pleistocene with different glacials is not warranted.
- l. 362-388 Same applies to the next paragraph. Tone things down, give alternatives like dispersals, drift etc.
- l. 404/5. Why not the alternatives?


Mike Thiv

---

## Round 0.2 · Minor Revisions

· Academic Editor

Minor Revisions

Dear authors

Thank you for submitting your manuscript to our journal. As you see our reviewers suggest a revision of your ms. If you are willing to do so, we would be happy to reconsider your revised manuscript.

Michael Wink

·

Basic reporting

see General Comments for the Author

Experimental design

see General Comments for the Author

Validity of the findings

see General Comments for the Author

Comments for the author

This is the second review of the paper by Kyrkou et al. analysing the phylogenetic relationships of Silene ciliata (= S.c.).

The ms. improved considerably in the discussion part and in several minor issues. As main point to consider I would still strongly advise to refrain from showing single cpDNA analyses (affecting fig. 3 a, b & c and fig. 5). The authors state in the response to reviewer 1 ‘that the figures were more difficult to interpret due to the large amount of information presented. We still believe that the haplotype networks are more informative if presented separately for each cpDNA region‘.
I think, however, that it does not make much sense: 1) The cp genome is inherited as a whole and 2) to my knowledge recombination in the cp DNA is not detected. 3) Following the total evidence approach it is better to combine the data (in case of no incongruence). Otherwise I could assume that the authors pick the marker-pattern arbitrarily depending on the outcomes.

As a second major point I recommend to restructure the MM part.
1) Studied species
2) Taxon selection (l. 132-137 + l. 143-144 + „other taxa selection“)
3) DNA markers (Study genome + l. 140-143)
4) DNA extr., amplification
5) Data analyses (incl. Alignment + „genetic analyses“)

In the discussion, the authors favour vicariance over dispersal. There are indeed some arguments for this, but the alternative could be discussed, too. I still think that the further discussion of the orographic structures as isolation and the “Pyrenees case” is speculative, but the authors marked it as such.

Minor points:
- Abstract: delete “in an attempt” l. 3; I would omit the last sentence “…hybridization .. mutational..” Since there are no data to support this. Instead a sentence that the evolution chromosome no. is not reflected in the cpDNA pattern might be useful and interesting.
- l. 101 => hypothesize
- l. 111, Studied
- l. 304 A RELATIVELY ancient
- l. 333 since there were Nunataks in the Alps just delete “completely”
- l. 378 substitute communication with exchange


Mike Thiv

---

## Round 0.3 · accepted · Accept

· Academic Editor

Accept

Dear authors

Thank you for resubmitting your manuscript. We are happy to accept your ms - thank you for submitting your research results to our journal.

Michael Wink